# Multitask Learning for Face Forgery Detection: A Joint Embedding Approach

## Abstract

Multitask learning for face forgery detection has experienced impressive successes in recent years. Nevertheless, the semantic relationships among different forgery detection tasks are generally overlooked in previous methods, which weakens knowledge transfer across tasks. Moreover, previously adopted multitask learning schemes require human intervention on allocating model capacity to each task and computing the loss weighting, which is bound to be suboptimal. In this paper, we aim at automated multitask learning for face forgery detection from a joint embedding perspective. We first define a set of coarse-to-fine face forgery detection tasks based on face attributes at different semantic levels. We describe the ground-truth for each task via a textural template, and train two encoders to jointly embed visual face images and textual descriptions in the shared feature space. In such a manner, the semantic closeness between two tasks is manifested as the distance in the learned feature space. Moreover, the capacity of the image encoder can be automatically allocated to each task through end-to-end optimization. Through joint embedding, face forgery detection can be performed by maximizing the feature similarity between the test face image and candidate textual descriptions. Extensive experiments show that the proposed method improves face forgery detection in terms of generalization to novel face manipulations. In addition, our multitask learning method renders some degree of model interpretation by providing human-understandable explanations.

## 1 Introduction

The emergence of deep generative models [1, 34, 67, 71] has significantly simplified and automated the process of generating realistic counterfeit face images, popularly known as DeepFake. The prevalence of falsified face images can erode the reliability and credibility of digital visual information. Additionally, the exploitation and manipulation of such technologies pose a threat to individual rights and national security.

Traditional DeepFake detectors were largely influenced by classic photo forensics [21] to expose forgery traces by examining statistical anomalies [51, 58], visual artifacts [32, 46, 50, 51, 59], and physical and geometric inconsistencies [15, 33, 35, 56]. With the rapid development of deep learning, there has recently been a growing consensus on exploiting multitask learning for face forgery detection [8, 10, 19, 41, 55, 80, 81]. The underlying assumption is that the primary task (*i.e.*, global face forgery classification) is likely to benefit from other highly relevant auxiliary tasks through knowledge transfer. Representative auxiliary tasks include manipulation type (and degree) classification [10], manipulation parameter estimation [75], blending boundary detection [41], spatial forgery localization [28], face reconstruction [8], and face segmentation [55].

The prevailing multitask learning paradigm for face forgery detection follows a discriminative approach, predicting multiple target outputs, one for each task, directly from the input face image. Such a paradigm suffers from two main drawbacks. First, semantic relationships across tasks are overlooked, which weakens knowledge transfer. For example, irrelevant information (*e.g.*, every detail of the face image in face reconstruction [8]) may be transferred across tasks. Second, extensive human expertise should be involved, when determining task-agnostic (and task-specific) model parameters and the loss weightings.

In this paper, we explore multitask learning for face forgery detection from a joint embedding perspective [38]. In the joint embedding architecture, both the input and the target output are encoded into latent representations in the shared feature space such that the irrelevant information can be discarded from feature encoding. More importantly, the semantic closeness between two tasks can be naturally modeled as the distance in the learned feature space, which is subsequently end-to-end optimized to facilitate knowledge transfer across multiple tasks. Meanwhile, joint embedding gives us a great opportunity to automate multitask learning in terms of allocating model capacity (*i.e.*, specifying task-agnostic and task-specific model parameters). In the context of face forgery detection, the parameters of the face image encoder are shared across all tasks, whose capacity is dynamically adjusted through end-to-end optimization. In addition, the multitask loss weightings can be automatically computed in either theoretical [45, 65] or empirical [13, 36, 47] ways.

More concretely, we first introduce three coarse-to-fine face forgery detection tasks based on face attributes at different semantic levels. Leveraging the recent advances in vision-language correspondence as joint embedding [61], we encode the binary labels of the three tasks via textural prompts, and thus the semantic dependencies among tasks can be represented with the textual embeddings in the representation space. Fig. 1 shows an example, in which we describe a fake face image with a set of coarse-to-fine textual descriptions: 1) "*A photo of a fake face,*" 2) "*A photo of a face with the global attribute of expression altered,*" and 3) "*A photo of a face with the local attribute of mouth altered.*" By jointly embedding the face image and all its associated textural prompts through a popular vision-language model - CLIP [61], face forgery detection can then be performed by maximizing the vision-language correspondence.

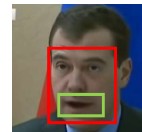

(1) A photo of a fake face
(2) A photo of a face with the global attribute of expression altered
(3) A photo of a face with the local attribute of mouth altered

Figure 1: Illustration of a fake face image with its textural descriptions of three coarse-to-fine face forgery detection tasks at different semantic levels.

**Our contributions** are threefold. First, we formulate multitask face forgery detection from a joint embedding perspective. Second, we define a set of coarse-to-fine face forgery detection tasks with corresponding textural templates to describe (fake) face images. Compared to previous multitask learning schemes, our instantiation gives rise to a more interpretable face forgery detector. Third, we conduct extensive experiments on five popular face forgery detection datasets, and show that our method performs favorably against state-of-the-art (SOTA) detectors in terms of generalization to novel face manipulations.

## 2 Related Work

In this section, we briefly review the literature on face forgery detection, multitask learning, and joint embedding architectures.

### 2.1 Face Forgery Detection

Many face forgery detection methods usually explore the specific clues to detect the forgery inspired by the traditional photo forensics [15, 32, 33, 35, 46, 50, 51, 56], in which they detect eye blinking [42], head pose [77], pupil shape [24], lipreading [26], statistical anomalies [43, 60, 66, 81], corneal specularity [29], and idiosyncratic behavioral patterns of a well-known person [3]. In recent years, there is a growing consensus of exploiting multitask learning on face forgery detection [8, 10, 41, 55, 81]. Besides the main face forgery classification task, these methods include auxiliary tasks to get performance improvement by knowledge transfer across tasks, such as manipulation type (and degree) classification [10], manipulation parameter estimation [75], blending boundary detection [41], spatial forgery localization [28], face reconstruction [8], and face segmentation [55].

With the development of deep learning, some advanced networks are employed to facilitate the face forgery detection based on multiple tasks, such as two-stream CNN [82], self-attention model [80], and vision transformers [19]. Additionally, more advanced training strategies are also utilized to enhance the forgery detectors, including adversarial learning [10], reconstruction learning [8], and meta learning [11]. However, the previous learning paradigm and human intervention are sub-optimal for multitask learning on face forgery detection. In this paper, we explore an automated multitask learning method for face forgery detection from the joint embedding perspective, where multiple tasks are encoded into the language prompts, and vision-language correspondence is transferred across tasks as the primary knowledge.

## 2.2 Multitask Learning

Multitask learning aims to jointly learn multiple related tasks to improve the generalization performance of all tasks by leveraging the knowledge contained in each [79]. Two main groups are model parameter sharing and loss weighting. The former involves both manual specifications of shared parameters [4, 22, 37, 54] and learning to determine parameters for specific tasks [52, 64, 68, 74]. Loss weighting is typically divided as follows: Pareto Optimization (PO) methods and weight adaption methods. PO methods formulate multitask learning as a multi-objective optimization [45, 65], and find a Pareto stationary solution for the optimal loss weighting. Weight adoption methods adaptively adjust the loss weights during training based on pre-defined heuristics, such as uncertainty [36], gradient normalization [13], and loss descending rate [47]. In this paper, we consider multitask learning from the joint embedding perspective, in which the semantic closeness between tasks can be manifested as the distance in the learned feature space. Moreover, we assume all parameters in the image encoder are shared, whose capacity is dynamically allocated to each task during end-to-end optimization. We also adopt the method in [47] for dynamic loss weighting.

## 2.3 Joint Embedding Architectures

Joint embedding architectures (JEA) [38] aim at learning to output similar embeddings for compatible inputs, $x$ and $y$, and dissimilar embeddings for incompatible inputs, which is different from the discriminative approaches that predict $y$ directly from $x$. Becker *et al.* [6] propose the first JEA for maximizing mutual information between representations from two views of the same scene. Later on, Bromley *et al.* [7] propose a contrastive method of JEA for signatures verification. After a long hiatus, JEA has been re-explored in face verification [14] and recognition [69], dimensionality reduction [25], and video feature learning [70]. With the emergence of self-supervised learning, the use of JEA has explored in recent years with methods training on contrastively (*e.g.*, PIRL [53], MoCo [27], and SimCLR [12]) or non-contrastively (*e.g.*, BYOL [23], Barlow Twins [78], and I-JEPA [5]). More recently, the emerging vision-language foundation models [30, 61] can also be grouped into JEA, in which two separate encoders encode the compatible visual (*i.e.*, $x$) and textual (*i.e.*, $y$) inputs into similar embeddings and contrast incompatible visual and textual embeddings. In this paper, we use CLIP [61], a joint vision-language model pretrained on massive image-text pairs, to implement the JEA to aid DeepFake detection by vision-language correspondence in the embedding space. Moreover, we end-to-end fine-tune the CLIP in the context of automated multitask learning.

# 3 Method

In this section, we present multitask learning for face forgery detection using a joint embedding approach, including preliminaries of the problem formulation, language prompts over multiple tasks, and specifications of loss functions. The main joint embedding framework for face forgery detection is shown in Fig. 2.

## 3.1 Preliminaries

Given a face image $x \in \mathbb{R}^N$, a face forgery detector $f_\theta : \mathbb{R}^N \mapsto \mathbb{R}$ aims to predict a binary label $y$ for the authenticity of $x$, *i.e.*, 0 as the real or 1 as the fake. Considering that existing forged face images are mainly generated by modifying face components/attributes, we include two other related tasks - global face manipulation detection and local face manipulation detection. We consider three

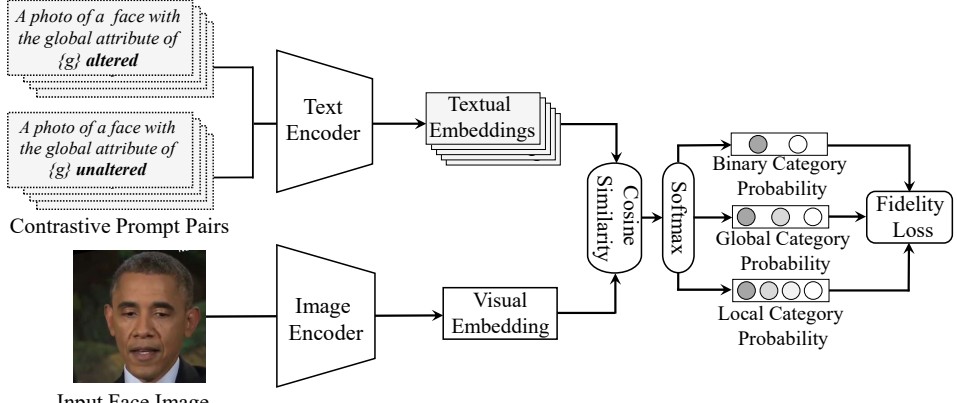

Figure 2: Proposed joint embedding paradigm for multitask face forgery detection.

face attributes (*i.e.*, expression, identity, and physical consistency[1]) for global face manipulations, and four face attributes (*i.e.*, eye, illumination, mouth, and nose) for local face manipulations. Notably, a face image may contain multiple attribute labels.

## 3.2 Multitask Language Prompts

For each face attribute label from multiple tasks, we encode the ground-truth labels via language prompts. In specific, we design textual templates as follows. 1) **binary level**: *a photo of a {c} face*, where $c \in \mathcal{C} = \{$real, fake$\}$; 2) **global-attribute level**: *A photo of a face with the global attribute of {g} altered*, where $g \in \mathcal{G} = \{$expression, identity, physical consistency$\}$; and 3) **local-attribute level**: *A photo of a face with the local attribute of {l} altered*, where $l \in \mathcal{L} = \{$eye, illumination, mouth, nose$\}$. Inspired by contrastive methods [27, 53] in the joint embedding architecture, we also introduce contrastive language prompts, which are opposite in meaning to the original textual templates. Thus, we can have a contrastive prompts pair for each attribute label, as follows: **global-attribute level**: $\{$(1) *A photo of a face with the global attribute of {g} altered*, (2) *A photo of a face with the global attribute of {g} unaltered*$\}$; **local-attribute level**: $\{$(1) *A photo of a face with the local attribute of {l} altered*, (2) *A photo of a face with the local attribute of {l} unaltered*$\}$. Notably, the binary level prompts naturally have the property of contrastive prompt pairing. In this way, multiple tasks are encoded into a text corpus $\mathcal{T}$, where each language prompt represents a ground-truth label $y$ of the corresponding task, and their semantic closeness can be learned through joint embedding.

## 3.3 Multitask Learning via Joint Embedding

**Joint Embedding Formulation**. Given the input face image $\boldsymbol{x}$ and the set of possible outputs $\mathcal{Y}$, we predict the output by minimizing an energy-based model [39], *i.e.*, $\hat{y} = \arg\min_{y \in \mathcal{Y}} E(\boldsymbol{x}, y)$, in the joint embedding architecture. In this paper, we construct $E$ by two encoders: one image encoder $\boldsymbol{f_\phi} : \mathbb{R}^N \mapsto \mathbb{R}^K$ for encoding the face image and one text encoder $\boldsymbol{g_\varphi} : \mathcal{T} \mapsto \mathbb{R}^K$ for encoding the language prompts, parameterized by $\boldsymbol{\phi}$ and $\boldsymbol{\varphi}$, respectively.

The ideal energy landscape of joint embedding satisfies that the energy is low for similar embeddings of compatible inputs, while energy is high for dissimilar embeddings [39]. Thus, we calculate the probability of similarity $\hat{p}(\cdot|\boldsymbol{x})$ between the visual embedding and textual embeddings for the following optimization. Let $\boldsymbol{u} \in \mathbb{R}^K$ be the visual embedding, and let $\boldsymbol{v} \in \mathbb{R}^K$ and $\bar{\boldsymbol{v}} \in \mathbb{R}^K$ be the textual embeddings from the two prompts opposing in meaning, we then estimate $\hat{p}(\cdot|\boldsymbol{x})$ as

$$\hat{p}(\cdot|\boldsymbol{x}) = \frac{1}{1 + e^{-(s-\bar{s})}}, \tag{1}$$

where

$$s = \frac{\langle \boldsymbol{u}, \boldsymbol{v} \rangle}{\|\boldsymbol{u}\|\|\boldsymbol{v}\|} \quad \text{and} \quad \bar{s} = \frac{\langle \boldsymbol{u}, \bar{\boldsymbol{v}} \rangle}{\|\boldsymbol{u}\|\|\bar{\boldsymbol{v}}\|}. \tag{2}$$

---

[1]We refer the interested readers to the Appendix for the detailed explanations.

$\langle \cdot, \cdot \rangle$ denotes the inner product and $\| \cdot \|$ represents the $\ell_2$-norm. The probability $\hat{p}(\cdot|\boldsymbol{x})$ is the abbreviation of $\hat{p}(c|\boldsymbol{x})$, $\hat{p}(g|\boldsymbol{x})$, and $\hat{p}(l|\boldsymbol{x})$ according to a specific task, and a larger probability indicates a closer match to the corresponding semantic meaning of $\boldsymbol{v}$.

**Losses for Multitask Learning**. We use the statistical distance measure in the form of fidelity loss [73] to calculate the losses for multitask learning. Given the predicted category probability $\hat{p}(c|\boldsymbol{x})$, we design the loss at the **binary level** as

$$\ell_1(\boldsymbol{x}; \boldsymbol{\theta}) = 1 - \sqrt{p(c|\boldsymbol{x})\hat{p}(c|\boldsymbol{x})} - \sqrt{(1 - p(c|\boldsymbol{x}))(1 - \hat{p}(c|\boldsymbol{x}))}, \qquad (3)$$

where $\boldsymbol{\theta} = \{\boldsymbol{\phi}, \boldsymbol{\varphi}\}$ indicates the learnable parameters in image and language encoders, and $p(c|\boldsymbol{x}) = 1$ if $\boldsymbol{x}$ belongs to the $c$ category or otherwise we have $p(c|\boldsymbol{x}) = 0$. In our setting, a face image can be assigned with labels regarding one or more global face attribute manipulations, which forms a typical multi-label classification problem. Therefore, the averaged loss at the **global-attribute level** can be defined as follows,

$$\ell_2(\boldsymbol{x}; \boldsymbol{\theta}) = \frac{1}{|\mathcal{G}|} \sum_{g \in \mathcal{G}} \left( 1 - \sqrt{p(g|\boldsymbol{x})\hat{p}(g|\boldsymbol{x})} - \sqrt{(1 - p(g|\boldsymbol{x}))(1 - \hat{p}(g|\boldsymbol{x}))} \right), \qquad (4)$$

where $p(g|\boldsymbol{x}) = 1$ if $\boldsymbol{x}$ belongs to the $g$ category, otherwise we have $p(g|\boldsymbol{x}) = 0$. Since the manipulations over different local face attributes may appear in one face image, we also consider it as a multi-label classification task, and the loss at the **local-attribute level** is:

$$\ell_3(\boldsymbol{x}; \boldsymbol{\theta}) = \frac{1}{|\mathcal{L}|} \sum_{l \in \mathcal{L}} \left( 1 - \sqrt{p(l|\boldsymbol{x})\hat{p}(l|\boldsymbol{x})} - \sqrt{(1 - p(l|\boldsymbol{x}))(1 - \hat{p}(l|\boldsymbol{x}))} \right), \qquad (5)$$

where $p(l|\boldsymbol{x}) = 1$ if $\boldsymbol{x}$ belongs to the $l$ category.

Given a minibatch of training data $\mathcal{B}$ at the $t$-th iteration, we evaluate the overall loss function via the weighted sum of the individual losses in different levels as follows,

$$\ell(\mathcal{B}, t; \boldsymbol{\theta}) = \frac{1}{|\mathcal{B}|} \sum_{\boldsymbol{x} \in \mathcal{B}} \left( \lambda_1(t)\ell_1(\boldsymbol{x}; \boldsymbol{\theta}) + \lambda_2(t)\ell_2(\boldsymbol{x}; \boldsymbol{\theta}) + \lambda_3(t)\ell_3(\boldsymbol{x}; \boldsymbol{\theta}) \right). \qquad (6)$$

Here, the weighting vector $\boldsymbol{\lambda}(t) = [\lambda_1(t), \lambda_2(t), \lambda_3(t)]^{\mathsf{T}}$ at the $t$-th iteration is automatically computed according to the relative descending rate [47]:

$$\lambda_i(t) = \frac{3 \exp\left(w_i(t-1)/\tau\right)}{\sum_{j=1}^{3} \exp\left(w_j(t-1)/\tau\right)}, \text{ where } w_i(t-1) = \frac{\ell_i(t-1)}{\ell_i(t-2)}, \qquad (7)$$

and $\tau$ is a fixed temperature parameter.

# 4 Experiments

## 4.1 Experimental Setup

**Datasets.** We adopt the widely used FF++ [63] dataset for training. It contains $1,000$ real videos, among which $720$ and $140$ are used for training and validation, respectively, and the remaining $140$ are reserved for testing. All videos are manipulated by four face forgery methods, including Deepfakes [1], Face2Face [72], FaceSwap [2], and NeuralTextures [71], with three compression levels, *i.e.*, no compression (denoted as Raw), slight compression with quantization parameter QP $= 23$ (denoted as C23), and severe compression with QP $= 40$ (denoted as C40). Following [10, 11, 26], C23 version is adopted by default in our experiments. We evaluate the generalizability of the proposed method on four popular DeepFake benchmarks, including FaceShifter (FSh) [40], Celeb-DF (CDF) [44], DeeperForensics-1.0 (DF-1.0) [31], and DeepFake Detection Challenge (DFDC) [18].

**Implementation Details.** To facilitate the multitask learning via joint embedding paradigm, we need face images associated with the proposed textual templates. In this paper, we adopt FF++ [63] to enrich the training data. Following the general generation procedures (*i.e.*, detecting face and then blending two faces according to the region-of-interest mask) in [10, 41], we focus on supplementing the tampering of "expression" on "eye" and individual face attribute that is linked to "physical consistency", *i.e.*, "eye", "illumination", "mouth", and "nose". Face attribute manipulations associated with other textual prompts are already included in FF++.

As for face pre-processing, we use RetinaFace [17] to detect faces and save the aligned face images as input with a size of $317 \times 317$. As in [63], we only extract the largest face and use an enlarged crop, $1.3\times$ the tight crop produced by the face detector.

As for the training, we use CLIP [61] to implement the joint embedding architecture, where we adopt ViT-B/32 [20] as the visual encoder and GPT-2 [62] with a base size of 63M-parameter as the text encoder. We then train the model by minimizing the loss using AdamW [49] with a decoupled weight decay of $1 \times 10^{-3}$. The initial learning rate is set to $1 \times 10^{-7}$, which changes following a cosine annealing schedule [48]. The model is optimized for 36 epochs with mini-batches of 32. Data augmentation strategy is also applied during training, which is a common trick in the face forgery detection [41, 76, 80], and details can be found in Sec. 4.3. A single NVIDIA RTX 3090 GPU is used during training.

## 4.2 Comparison with SOTA Methods

We compare our method with the several SOTA methods, including Face X-ray [41], PCL [81], MADD [80], LipForensics [26], RECCE [8], SBI [66], ICT [19], SLADD [10], and OST [11], to demonstrate its superiority. The test performance on five datasets are listed in Table 1. Table 1 shows that many methods do not perform satisfactorily on face forgery detection, while the proposed method outperforms all the recent SOTA, achieving $92.33\%$ of AUC averaged from five test datasets and surpassing the second best, *i.e.*, LipForensics, by $2.79\%$ in the term of Mean AUC over datasets including FF++ [63]. For

Table 1: **Comparison results with the SOTA**. All models are developed using the training set of FF++ (or its augmented versions) and tested on the test set of FF++ and other four independent datasets. The evaluation metric we adopt is AUC (%). In the last column are the mean AUC numbers over datasets including / excluding the FF++ test set to emphasize cross-dataset generalization performance. The best results are highlighted in bold.

| Method | FF++ | CDF | FSh | DF-1.0 | DFDC | Mean AUC |
|---|---|---|---|---|---|---|
| Face X-ray [41] | 98.37 | 80.43 | 92.80 | 86.80 | 65.50 | 84.78 / 81.38 |
| PCL [81] | 99.11 | 81.80 | – | **99.40** | 67.50 | 86.95 / 82.90 |
| MADD [80] | 98.97 | 77.44 | 97.17 | 66.58 | 67.94 | 81.62 / 77.28 |
| LipForensics [26] | **99.90** | 82.40 | 97.10 | 97.60 | 73.50 | 89.54 / 87.65 |
| RECCE [8] | 99.32 | 68.71 | 70.58 | 74.10 | 69.06 | 76.35 / 70.61 |
| SBI [66] | 99.64 | **93.18** | 97.40 | 77.70 | 72.42 | 88.07 / 85.18 |
| ICT [19] | 90.22 | 85.71 | 95.97 | 93.57 | 76.74 | 88.44 / 88.00 |
| SLADD [10] | 98.40 | 79.70 | – | 77.80 | 76.05 | 82.99 / 77.85 |
| OST [11] | 98.20 | 74.80 | – | 93.08 | 77.73 | 84.95 / 81.87 |
| Ours | 98.49 | 89.02 | **98.68** | 93.38 | **82.06** | **92.33 / 90.79** |

cross-dataset generalizability comparison, the proposed method also surpasses the second best (*i.e.*, ICT) and third best (*i.e.*, LipForensics) by $2.79\%$ and $3.14\%$, respectively. In addition, we also have several interesting observations. **First**, all the methods can achieve saturated performance in FF++ [63], while underperform in the rest datasets, such as CDF [44] and DFDC [18]. This suggests that the forgery cues in FF++ are easier to spot and overfit by these forgery detectors. **Second**, SBI reports a very high AUC of $93.18\%$ on CDF, while performing unsatisfactorily on DF-1.0 [31] and DFDC. Similar results are also demonstrated by PCL, which exhibits an exceedingly high AUC of $99.40\%$ on DF-1.0 but underperforms in DFDC. This may arise due to the overfitting on the low-level features, such as statistical inconsistency (*e.g.*, landmark and color mismatch). **Third**, all methods obtain relatively low scores on DFDC, which we attribute to the domain shift caused by significantly different filming conditions. However, our method achieves a relative satisfactory result with a score of $82.06\%$, surpassing the second best by $4.33\%$. In summary, the remarkable results validate the effectiveness and superiority of the proposed joint-embedding-based multitask learning for DeepFake detection.

## 4.3 Robustness Analysis

In this subsection, we study the robustness performance of the proposed method. Following [31], we consider four popular perturbations (*i.e.*, Patch Substitution (Patch-Sub), additive white Gaussian Noise contamination (Noise), Gaussian Blurring (Blur), and pixelation), and only four severity levels

Table 2: **Robustness results to low-level image perturbations**, including patch substitution (Patch-Sub), Gaussian noise contamination (Noise), Gaussian blurring (Blur), and pixelation. We constrain the robustness evaluation on the perturbation levels that do not noticeably distort the main face semantics.

| Method | Clean AUC | Patch-Sub | Noise | Blur | Pixelation | Mean AUC | Drop Rate |
|---|---|---|---|---|---|---|---|
| Face X-ray [41] | 98.37 | **97.72** | 51.13 | 88.98 | 92.33 | 82.54 | -16.09% |
| CNND [76] | 99.56 | 96.25 | 57.25 | 92.61 | 90.10 | 84.05 | -15.58% |
| LipForensics [26] | 99.90 | 88.63 | 80.00 | **96.62** | **96.63** | **90.47** | -9.44% |
| Ours (w/o Aug) | 98.66 | 92.47 | 73.12 | 55.20 | 57.17 | 69.49 | -29.57% |
| Ours | 98.49 | 97.65 | **82.85** | 87.31 | 90.70 | 89.63 | **-8.99**% |

(*i.e.*, from level 1 to level 4) are considered in the experiments[2]. Two different models are evaluated in this section, *i.e.*, our model training without data augmentation (denoted as Ours (w/o Aug)) and our model training with data augmentation strategy (denoted as Ours). In specific, when training with data augmentation strategy, each training data is augmented with a probability of 0.3 by one randomly chosen perturbation during training, in which severity level is randomly applied at level 1 or 2.

To begin, we first evaluate the robustness for the model without data augmentation. We find that the CLIP-based model is sensitive to the perturbations to images, which we argue that the vision-language correspondence is corrupted by perturbations. We then evaluate the model training with data augmentation. In Table 2, we find that training with a slight data augmentation can alleviate the model sensitivity to the perturbations, and achieve a satisfactory performance on average. Moreover, the model of Ours also maintains a satisfactory performance on pixelation and Blur. It is noteworthy that CNND [76] and Face X-ray [41] also augment their training data by compression and blurring during training, thus leading to good robustness to perturbations of pixelation and Blur. Fig. 3 demonstrates the effect of increasing the severity for each perturbation, where we compare with Xception [63], CNND, PatchForensics [9], Face X-ray, and LipForensics [26]. It can be observed that the proposed method maintains a good performance against the perturbations by Patch-Sub and Noise, while other methods suffer from the Noise, and LipForensics also suffers from the Patch-Sub.

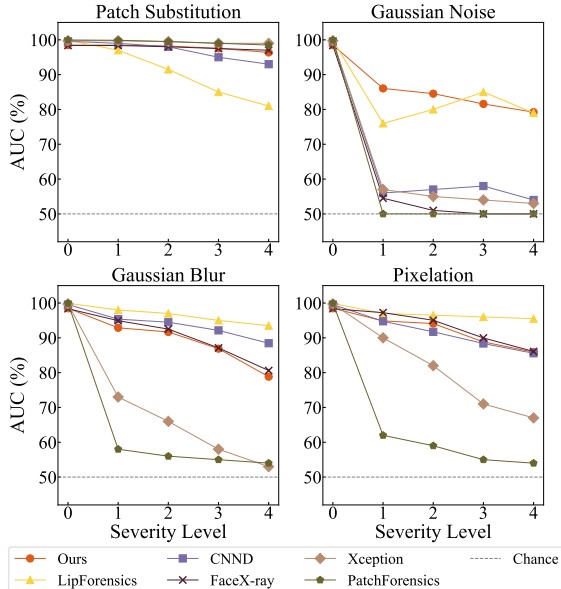

Figure 3: Robustness results in terms of AUC. Models are trained on the train set of FF++ and tested on perturbed test sets. Zoom in for clearer comparison.

### 4.4 Ablation Studies

**Joint Embedding Framework.** We conducted a series of ablations to verify the instantiated joint embedding framework by CLIP [61]. We first (1) evaluate the pretrained CLIP, and then (2) fine-tune it with the frozen text encoder on FF++ [63]. The following ablations adopt the same training procedure, while differing in two alternatives: (3) using equal task weights for multiple tasks instead of dynamic loss weighting; (4) training without the contrastive prompt pairs, *i.e.*, no contrastive textual descriptions are used during training. From Table 3, we can observe that freezing language encoder negatively affects the generalization performance, which we believe is because forgery-related concepts have not been sufficiently captured during the pretraining stage of CLIP. We also find that utilizing contrastive prompts can improve generalization, further indicating the contrasting

---

[2]The perturbations on severity level 5 often make the face semantically unrecognized, leading meaningless to detect its authenticity.

operation can benefit the joint embedding methods [12, 27]. Moreover, including the dynamic loss weighting scheme is advantageous as it not only yields a slight improvement compared to using equal task weights but also frees us from the burdensome task of hyper-parameter tuning.

**Textual Templates.** In this subsection, we investigate how the textual template design affects the model performance. We try three different alternatives from single task to three tasks: (5) binary-level text templates, *i.e.*, single task formulation only considering the label of real or fake; (6) two-level separate text templates, *i.e.*, two-level-task formulation, where we consider the separate templates describing the overall authenticity and global face attributes; and (7) the joint text templates putting together labels from three tasks, *e.g.*, "*A photo of a {fake} face with the global attribute of {expression} and the local attribute of {mouth} are altered*". The joint probability over multiple tasks can be computed from the similarities between the image embedding and all candidate textual embeddings. Then, we marginalize the joint distribution to obtain the marginal probability for each task. From Table 3, we can observe that the performance of the model using joint templates is inferior to that of the model using separate templates (*i.e.*, Ours (Default)), indicating that separate templates for each task are more conducive for learning the semantic closeness between two face forgery detection tasks in joint embedding. On the other hand, less tasks (*i.e.*, single task and two tasks) result in the inferior performance. Notably, benefiting from the joint embedding, the model using binary templates also achieves comparable results on generalization, though it only classifies the overall authenticity of the face.

Table 3: **Ablation Studies**. Baseline denotes the single-task formulation w/o contrastive textual pairing nor data augmentation, optimized for the BCE loss.

| Model Variant | CDF | FSh | DF-1.0 | DFDC | Mean AUC |
|---|---|---|---|---|---|
| (1) Pretrained CLIP | 65.38 | 51.04 | 53.38 | 55.56 | 56.34 |
| (2) Frozen $g_\varphi$ | 90.56 | 98.92 | 91.22 | 80.19 | 90.22 |
| (3) Equal Weights | 88.32 | 98.77 | 92.93 | 82.27 | 90.57 |
| (4) w/o Contrastive Pair | 87.89 | 98.34 | 93.30 | 81.27 | 90.20 |
| (5) Binary Templates | 85.03 | 98.42 | 93.33 | 81.58 | 89.59 |
| (6) Two-Levels | 87.57 | 98.47 | 93.74 | 80.81 | 90.15 |
| (7) Joint Templates | 88.05 | 98.42 | 94.21 | 81.31 | 90.50 |
| (8) ViT-B/16 | 88.13 | 99.62 | 93.30 | 82.30 | 90.84 |
| (9) ViT-L/14 | **90.78** | **99.95** | **98.60** | **86.22** | **93.89** |
| (10) BCE Loss | 86.45 | 98.35 | 93.40 | 80.81 | 89.75 |
| (11) Probabilistic Loss | 87.81 | 98.41 | 91.55 | 81.18 | 89.74 |
| Ours (Baseline) | 71.63 | 98.19 | 89.94 | 74.02 | 83.44 |
| Ours (w/o Aug) | 85.53 | 98.82 | 93.95 | 80.41 | 89.68 |
| Ours (Default) | 89.02 | 98.68 | 93.38 | 82.06 | 90.79 |

**Encoder Architecture.** In this subsection, we investigate other visual encoders with different settings and model sizes. In specific, we choose (8) ViT-B/16 [20] and (9) ViT-L/14 [20]. As shown in Table 3, two alternative ViT-based architectures achieve better results on generalization. However, the larger model will result in both computationally more expensive and time-consuming.

**Multitask Objective.** In this subsection, we study how different optimization objectives affect the performance. As a reference, we first replace the fidelity loss functions with (10) binary cross entropy loss (BCE Loss). We also adopt the (11) hierarchical probabilistic loss [16] to jointly formulate multi-level classification tasks under a hierarchical label semantic graph. The relative similarity score (*i.e.*, $s - \bar{s}$), as a raw score, for each node in the label hierarchy, will be converted into marginal probabilities for loss computation. From Table 3, we observed that the proposed method outperforms the variant trained with BCE loss, thus providing evidence for the effectiveness of the designed fidelity losses. Furthermore, Table 3 shows that fidelity loss yields better performance than the hierarchical probabilistic loss, suggesting that implicitly learning the semantic dependencies may be better than explicitly encoding the prior knowledge in the label hierarchy graph in advance.

### 4.5 Discussion: Vision-Language Correspondence

**Human-Understandable Interpretation**. The proposed joint embedding approach enjoys the vision-language correspondence, which naturally provides model interpretations by providing human-understandable explanations. Fig. 4 shows some examples of FF++ [63], in which Deepfakes [1] indicate the identity swap, leading all local parts of the face are fake; and NeuralTextures [71] modify the expression in the mouth part. Take an example of NeuralTextures, the texts with a probability over $50\%$ include "fake", "expression", and "mouth". Hence, we consider this face image to be fake

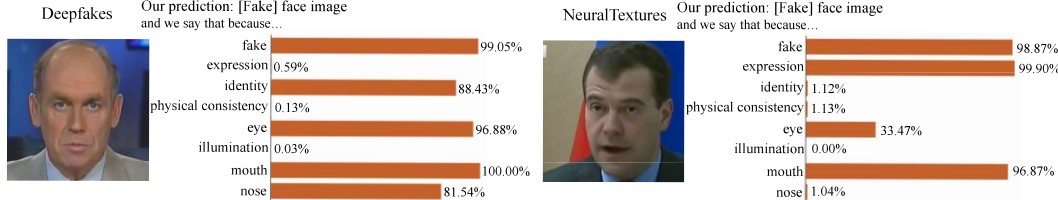

Figure 4: Bar charts of the similarity scores between the visual image and the textual descriptions a form of human-understandable explanations.

because the model's prediction relies on the following three textual prompts: "*a photo of a fake face*", "*a photo of a face with the global attribute of expression altered*", and "*a photo of a face with the local attribute of mouth altered*". More examples can be found in Appendix.

**Semantic Closeness across Tasks**. We show the semantic closeness across tasks by a correlation matrix in Fig. 5, in which each entry is represented by the cosine similarity between two textual embeddings from the language prompts depicting the specific tasks. From Fig. 5, we can observe that the text encoder of the pretrained CLIP has not sufficiently captured the semantic closeness across tasks and treats most tasks equally, further verifying the results of the variant with frozen text encoder in Table 3. After joint embedding learning on the forged faces, the semantic closeness across tasks can be sufficiently learned, *e.g.*, the concept of "identity" forgery is more related to the "nose", "mouth", and "eye", thus improving the performance of multitask learning for face forgery detection.

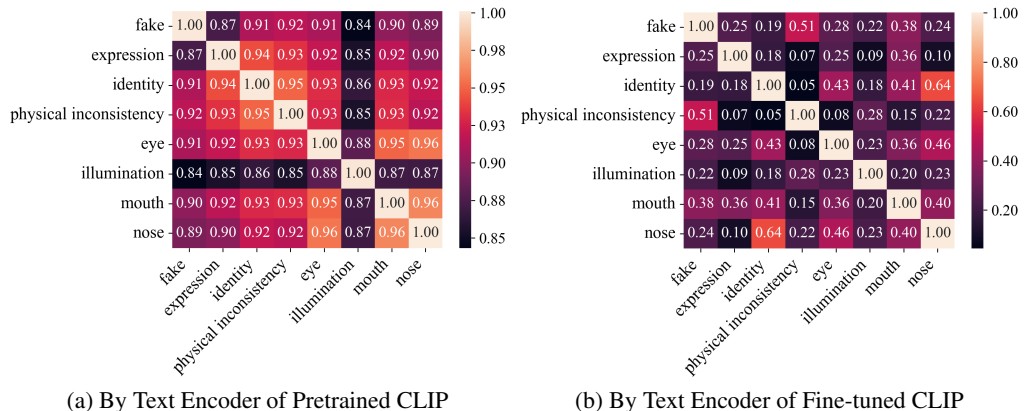

(a) By Text Encoder of Pretrained CLIP          (b) By Text Encoder of Fine-tuned CLIP

Figure 5: Illustration of semantic closeness across tasks before and after fine-tuning.

# 5   Conclusion and Limitations

**Conclusion**. In this paper, we consider multitask learning for face forgery detection from the joint embedding perspective. We have designed a set of coarse-to-fine language prompts to represent multiple tasks for face forgery detection. We then take an automated multitask learning scheme to train two encoders to joint embed visual face images and textual descriptions. Thus, semantic closeness across tasks is manifested as the distance in the learned feature space, thus improving multitask learning. From extensive experiments, vision-language correspondence after joint embedding shows great promise to support better face forgery detection by maximizing the feature similarity between the face image and candidate textual prompts, verifying the effectiveness and superiority of the proposed method. Moreover, the joint embedding scheme also renders some degree of model interpretation in a human-friendly way.

**Limitations**. The proposed method relies on the assumption that the forged faces are generated with the blending operation [41]. Thus, it may perform unsatisfactorily when fake face images are totally synthesized by GAN- or diffusion-model-based methods. Additionally, our model is image-based, though it can handle video-based DeepFake by sampling frames for prediction, it may fail when encountering the fake video manipulated by only lowering the frame rate [57].

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
