# Multitask Learning for Face Forgery Detection: A Joint Embedding Approach
# — Appendix —

## 1   Regarding the Manipulation of Physical Consistency

The physical inconsistency has been widely explored in photo forensics [7, 8, 11, 13, 18] and recently shed some light to face forgery detection, especially the utilization of illumination inconsistency [9]. We consider that the concept of "physical consistency" manipulation can also be used to examine whether different global/local regions of the face image during imaging come from the same 3D physical scene.

To avoid any conceptual confusion, we first distinguish it from "identity" manipulation. Let us consider naively swapping the faces of two **real** face images, and making an analysis. From the perspective of the background, we may conclude that the face identity has been altered; but from the perspective of the face itself, the background has changed (albeit still authentic, not artificially generated) and the authentic face is simply not present in its original background, resulting in physical inconsistency. In the context of face forgery detection, we prioritize the face as the primary object of interest and therefore adopt the second perspective, which emphasizes the importance of "physical consistency". In a typical scenario found in current datasets [6, 10, 14, 16, 20], a forged image consists of a real background and a fake face. In this case, focusing on the face as the main object of interest naturally falls under "identity" manipulation.

In this paper, we implement the manipulation of "physical consistency" as follows: 1) blending two real faces; 2) blending the local face part(s) (*e.g.*, "eye", "mouth", and "nose") from one real face to another person's face; and 3) introducing illumination inconsistency during face swapping of two real faces.

## 2   More Details of Experimental Setup

**Generation Details of Enriched Training Data.** We here introduce how to generate the enriched training data associated with the proposed textual templates based on FF++ [20]. Motivated by Face X-ray [15] and SLADD [5], we create the fake face through three steps: 1) given a real face image as the background, search for the nearest real face image as the foreground using face landmarks when dealing with "physical consistency" manipulations; otherwise, we directly use the corresponding fake image in FF++ as the foreground; 2) generate the

Table 1: **Illumination (in)consistency processing.** The symbol of "✓" means the illumination inconsistency processing, and "✗" signifies other physical inconsistency situations that may arise from blending two real faces or local face parts from one individual to another's face, while ensuring illumination consistency across the resulting image.

|  | w/ random brightness | w/o random brightness |
|---|---|---|
| w/ color correction | ✓ | ✗ |
| w/o color correction | ✓ | ✓ |

Submitted to 37th Conference on Neural Information Processing Systems (NeurIPS 2023). Do not distribute.

face mask from the convex hull of the face landmarks based on the background face; 3) blend two face images according to the region-of-interest mask, such as local eye region or the whole face region. Following Face X-ray[1], we adopt the soft mask, which is the binary mask after the Gaussian blur, when blending two images. Notably, for "illumination" manipulation, we apply some illumination-inconsistency operations during blending, such as applying random brightness (we implement it by using an image processing toolbox - Albumentations [1]) and/or no color correction [19]. Table 1 lists the specific operations, in which "✓" denotes the illumination-inconsistency processing combination, while "✗" is for the illumination-consistency operations. Besides simulating the illumination inconsistency, we always apply color correction when blending two faces based on the region-of-interest mask.

**Datasets.** We here introduce more details about four advanced datasets of DeepFake detection, *i.e.*, FaceShifter (FSh) [14], Celeb-DF (CDF) [16], DeeperForensics-1.0 (DF-1.0) [10], and DeepFake Detection Challenge (DFDC) [6].

FSh[2] is a published dataset containing 1,000 fake videos, which are generated by a more sophisticated face swapping technique, FaceShifter [14], based on the real videos from FF++. Therefore, FSh follows the same train/val/test splits as in FF++ and provides three subsets based on compression levels, *i.e.*, no compression (denoted as Raw), slight compression with quantization parameter $QP = 23$ (denoted as C23), and severe compression with $QP = 40$ (denoted as C40). Unless stated otherwise, C23 version is adopted by default in our experiments.

CDF dataset[3] is based on videos of celebrities, including 590 original videos collected from YouTube with subjects of different ages, ethnic groups, and genders, and $5,639$ corresponding DeepFake videos. CDF utilizes the improved DeepFake synthesis algorithm with more efforts on color match and temporal consistency, thus leading to a better visual quality of DeepFake videos. Further, we use the test set of the CDF for experiments.

DF-1.0[4] is a large-scale dataset, which contains more than 11,000 manipulated videos. The source videos are carefully collected on paid actors from different countries in a controlled scenario for better quality and diversity. All the manipulated videos are generated by DVAE [10], a newly proposed many-to-many end-to-end face swapping method considering temporal consistency. We use test split instructed in the dataset for experiments.

DFDC[5] dataset is a million-scale dataset and also one of the most challenging datasets for real-world face forgery detection. DFDC contains more than 100,000 videos produced with several DeepFake (*e.g.*, DeepFaceLab [2]), GAN-based (*e.g.*, StyleFAN [12], FSGAN [17], NTH [23]), and non-learned methods. In particular, DFDC provides a subset of $5,000$ videos for test, including $1,000$ real videos and $4,000$ fake videos. Unless stated otherwise, we use this test set by default in our experiments.

## 3 Additional Results on the Effect of Training Data Supplementary

In the proposed joint embedding approach for face forgery detection, we encode the ground-truth labels via a set of language prompts for each face attribute label from multiple tasks. To better leverage these language prompts, we introduce additional training data to compensate for the lack of vision-language correspondence in FF++ [20].

In this section, we explore the impact of training data supplementary on model performance. Table 2 demonstrates the re-

Table 2: **Additional Results on the Effect of Training Data Supplementary**. Baseline denotes the single-task formulation w/o contrastive textual pairing and data augmentation, optimized for the BCE loss.

| Model Variant | CDF | FSh | DF-1.0 | DFDC | Mean AUC |
|---|---|---|---|---|---|
| w/o DataSupp | 80.76 | 98.05 | 90.68 | 75.94 | 86.36 |
| Ours (Baseline) | 71.63 | 98.19 | 89.94 | 74.02 | 83.44 |
| Ours | 89.02 | 98.68 | 93.38 | 82.06 | 90.79 |

---

[1]https://github.com/AlgoHunt/Face-Xray

[2]https://github.com/ondyari/FaceForensics/tree/master/dataset/FaceShifter

[3]https://github.com/yuezunli/celeb-deepfakeforensics

[4]https://github.com/EndlessSora/DeeperForensics-1.0/tree/master/dataset

[5]https://ai.facebook.com/datasets/dfdc/

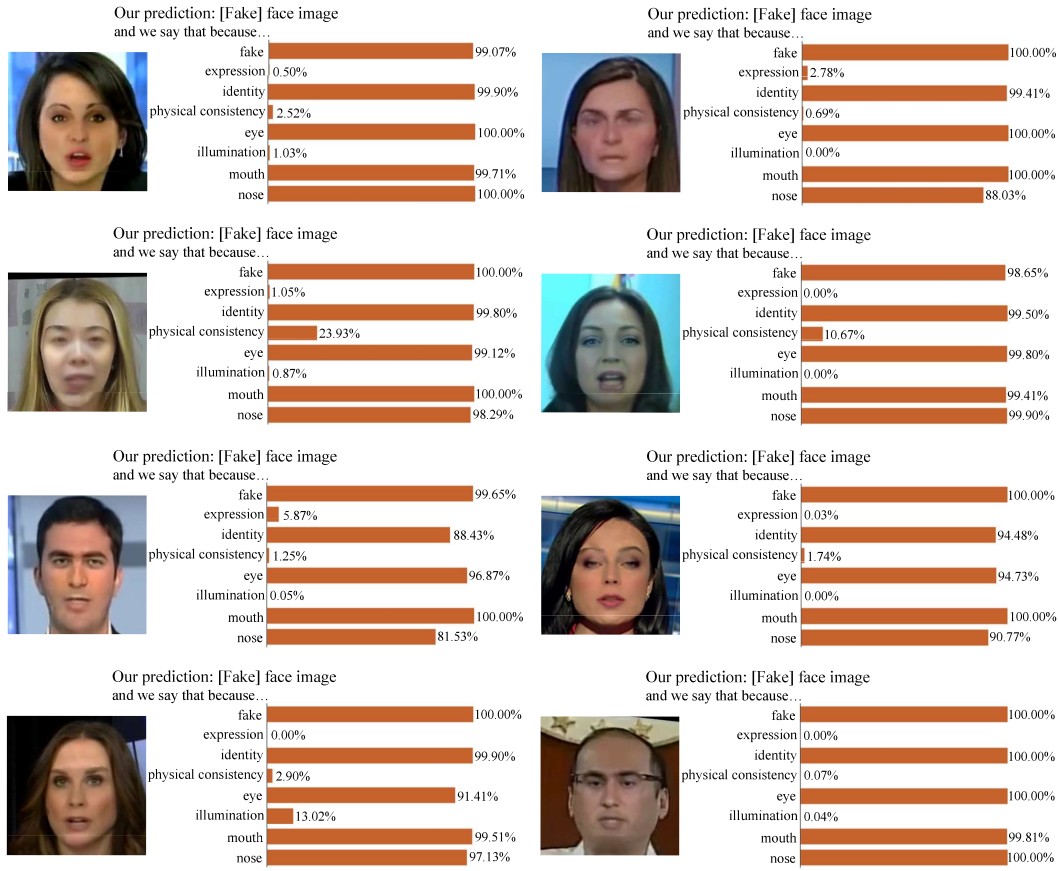

Figure 1: Bar charts of the similarity scores between the visual image and the textual descriptions. Face images are from the **Deepfakes** [3] subset in FF++ [20]. Zoom in for best view.

sults. From Table 2, we can observe that introducing additional face semantics data during training improves the model's ability on generalization, suggesting language prompts combined with appropriate visual data can fully take advantage of the joint embedding architecture for DeepFake detection, thus improving the performance of forgery detection.

## 4   Additional Vision-Language Correspondence Examples

In this section, we provide additional examples of bar charts of the similarity scores between the visual image and the textual descriptions, as illustrated in Fig. 1, Fig. 2, Fig. 3, and Fig. 4. All examples are obtained from the FF++ dataset [20], where Deepfakes [3] and FaceSwap [4] indicate the identity swap, leading all local parts (*i.e.*, eye, mouth, and nose) of the face are fake; and Face2Face [22] and NeuralTextures [21] modify the expression in the mouth part semantically.

## 5   Failure Cases

In this section, we provide examples of failure cases in Fig. 5 and Fig. 6, which can be divided into two categories in general: 1) misclassification of overall authenticity; and 2) misclassification of global/local face attributes.

**Misclassification of Overall Authenticity.** In general, we notice that poor visual quality (Fig. 5 (a)) or uneven local illumination (Fig. 5 (b)) can easily mislead the model to judge the real face image as fake, because these factors commonly appear in the process of face forgery process. In addition, in cases where the fake face images possess high visual quality and feature detailed facial

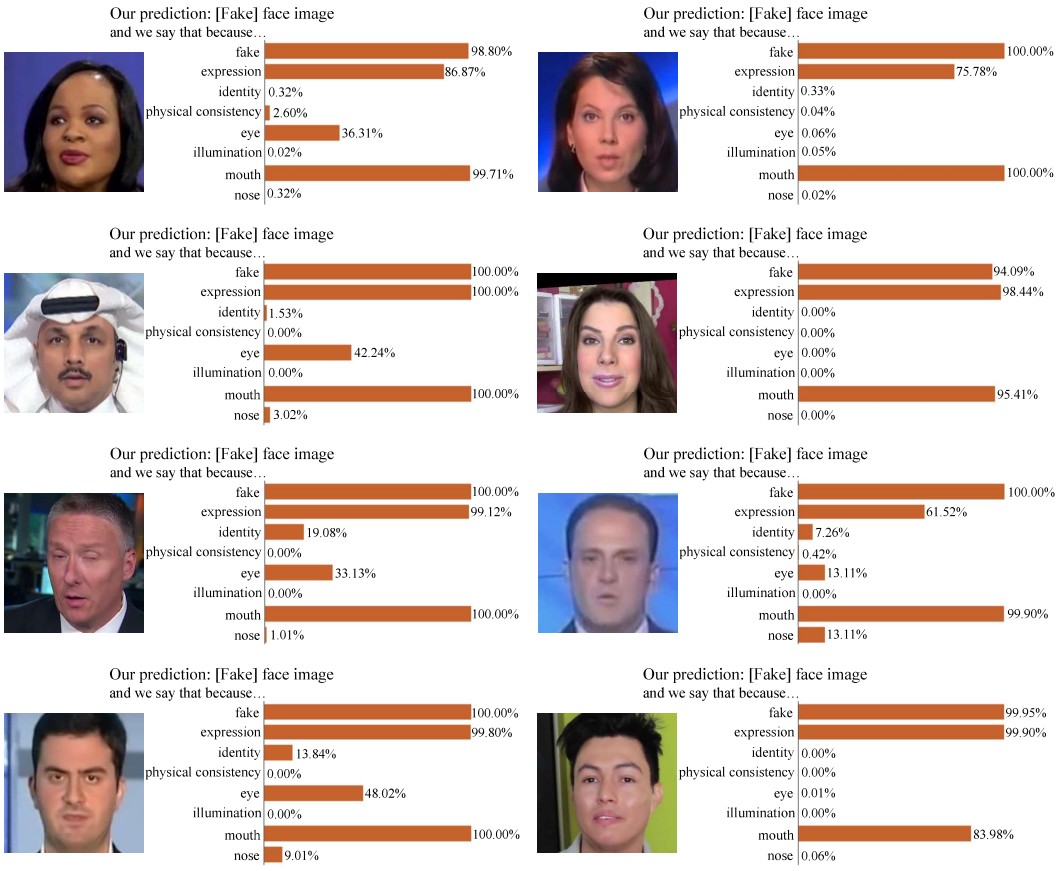

Figure 2: Bar charts of the similarity scores between the visual image and the textual descriptions a form of human-understandable explanations. Face images are from **Face2Face** [22] subset in FF++ [20]. Zoom in for best view.

components (Fig. 5 (c)-(d)), such as the eyes, mouth, and nose, the model may be deceived into incorrectly classifying these fake faces as authentic.

**Misclassification of Global/Local Face Attributes.** We here provide some typical failure examples when classifying each manipulation in FF++ [20], which are shown in Fig. 6. From Fig. 6, we can observe some several findings. **First**, when the target face and source face have different physical attributes (*e.g.*, hats, accessories, *etc*.), these physical attributes are also incorporated during the forgery generation process, resulting in severe artifacts and inconsistencies in the forged face (see left panel in Fig. 6 (b)), particularly in non-facial regions such as the forehead, that can mislead the model's prediction on specific face attributes. **Second**, mismatched landmarks between the target and source faces can cause distortions (*e.g.*, eyes and nose) in the generated fake face (see right panel in Fig. 6 (b)), leading the model to predict additional attribute label of "physical consistency". **Third**, parametric-face-model-based manipulations of Face2Face [22] may lead to imperfect artifacts similar to Deepfakes around the blending boundary and local face parts (see right panel in Fig. 6 (c)), thus leading to misclassification as identity change. **Fourth**, the poor visual quality is also an essential factor in deceiving the model to make incorrect predictions, such as the examples in Fig. 6 (d) for NeuralTextures [21].

Nonetheless, the proposed method prioritizes predicting the overall authenticity of face images rather than conducting multi-level classification on face forgeries. Hence, misclassifications of global/local face attributes are acceptable as long as the primary goal is achieved.

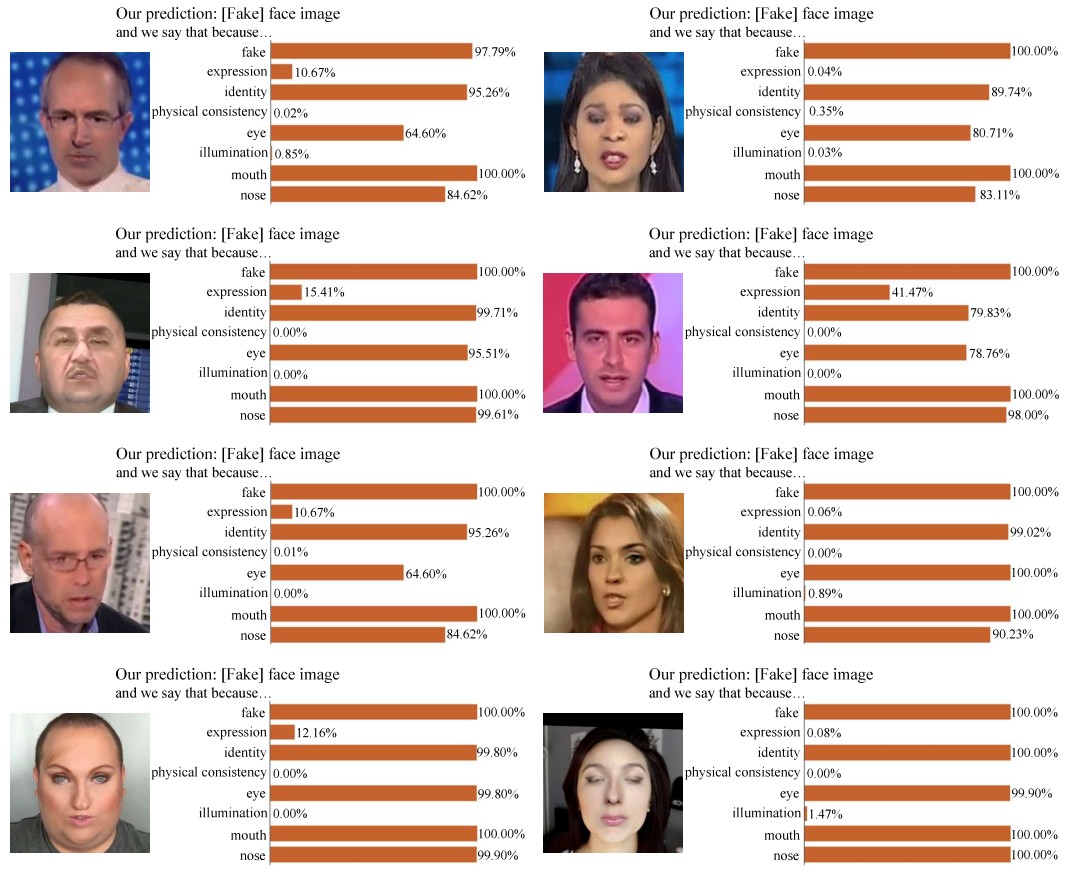

Figure 3: Bar charts of the similarity scores between the visual image and the textual descriptions a form of human-understandable explanations. Face images are from **FaceSwap** [4] subset in FF++ [20]. Zoom in for best view.