# OpenReview forum: "Multitask Learning for Face Forgery Detection: A Joint Embedding Approach"
_NeurIPS.cc/2023/Conference — Submitted to NeurIPS 2023_

### Official Review · Reviewer_7SFc · 2023-07-03

**Soundness:** 3 good
**Presentation:** 2 fair
**Contribution:** 3 good
**Rating:** 5
**Confidence:** 4

**Summary:**

The proposed method introduces a novel approach to deepfake detection by integrating natural language and image information. Moreover, it attains state-of-the-art performance on several contemporary deepfake datasets and can generate explanatory sentences that justify the authenticity or falsity of the input image, which is crucial in the field of deepfake detection.

**Strengths:**

See Questions Section in detail.

**Weaknesses:**

See Questions Section in detail.

**Questions:**

Although the proposed method demonstrates remarkable performance on contemporary deepfake datasets, the supplemental materials reveal that without the proposed dataSup scheme, the method can only achieve 80.76 and 75.94 AUC on Celeb-DF and DFDC, respectively, which is inferior to state-of-the-art methods.

Based on this, I have the following reservations about the submitted manuscript:

1. It would be preferable for the authors to compare the performance of their method with the proposed dataSup scheme and previous methods such as Face X-Ray, SBI and SLADD using a unified backbone and then analyze the impact of different dataSup schemes.
2. The data augmentation scheme is essential according to Table 2 in the supplemental materials. Therefore, I recommend that the authors elaborate on their dataSup scheme in detail as the current description of the scheme is obscure and difficult to replicate.

**Limitations:**

The limitations are well illustrated in the manuscript.

---

> ### Author Rebuttal · Authors · 2023-08-09
>
> **Q1. It would be preferable for the authors to compare the performance of their method with the proposed dataSup scheme and previous methods such as Face X-Ray, SBI and SLADD using a unified backbone and then analyze the impact of different dataSup schemes.**
>
> **A1:** Thanks for the excellent comment. As suggested by the reviewer, we provide more comparison results with a unified backbone but different dataSup schemes. The results in the following table further validate the effectiveness of the proposed dataSup scheme over those in Face X-Ray and SBI.
>
> |  DataSup Scheme | CDF  |  FSh |  DF-1.0 | DFDC  |  Mean AUC |
> | ------------ | ------------ | ------------ | ------------ | ------------ | ------------ |
> | Face X-Ray  |  84.34 | 98.36  | 91.12  | 77.88  | 87.93  |
> | SBI  | 87.39  |  98.31 | 92.09  | 79.02  | 89.20  |
> | Ours  | **89.02**  | **98.68**  |  **93.38**| **82.06** | **90.79**  |
>
> **Q2. The data augmentation scheme is essential according to Table 2 in the supplemental materials. Therefore, I recommend that the authors elaborate on their dataSup scheme in detail as the current description of the scheme is obscure and difficult to replicate.**
>
> **A2:** Thanks for the comment. We provide more details as follows, and we will also make the source code regarding the dataSup scheme publicly available for reference.
>
> **Expression-Eye**. (i) Choose **fake** data from Deepfakes and FaceSwap on FF++ with larger facial and expression modifications, particularly in the eye region, compared to the original faces.
> (ii) Given a real face on FF++ as the background face, we directly use the chosen fake face image on FF++ to supply the fake eye part(s) as the foreground;
> (iii) Generate the region-of-interest mask, i.e., the mask of the eye(s), based on the background face landmarks;
> (iv) Apply the color correction on the foreground;
> (v) Blend the background and the foreground according to the region-of-interest mask, in which we follow Face X-ray to adopt the Gaussian blurred binary mask.
>
> **Physical inconsistency-Eye/Mouth/Nose**. (i) Given a real face on FF++ as the background face, we search for the nearest **real** face images (excluding the real faces with the same ID) as the foreground to provide the local fake part(s);
> (ii) Generate the region-of-interest mask, i.e., the mask of the eye(s), mouth, or nose, based on the background face landmarks;
> (iii) Apply the color correction on the foreground;
> (iv) Blend the background and the foreground according to the region-of-interest mask.
>
> **Physical inconsistency-illumination.** (i) Given a real face on FF++ as the background face, we search for the nearest **real** face images (excluding the real faces with the same ID) as the foreground face;
> (ii) Generate the whole face mask based on the face landmarks of the background face;
> (iii) Apply illumination inconsistency operation on the foreground face, in which we have three alternatives: 1) random brightness + color correction, 2) random brightness, and 3) no correction. Detailed combinations are listed in Table 1 of the Appendix;
> (iv) Blend the background and the foreground according to the whole face mask.
>
> **Physical inconsistency**. (i) Given a real face on FF++ as the background face, we search for the nearest **real** face images (excluding the real faces with the same ID) as the foreground face;
> (ii) Generate the whole face mask based on the face landmarks of the background face;
> (iii) Apply the color correction on the foreground face;
> (iv) Blend the background and the foreground according to the region-of-interest mask.

---

> > ### Comment · Reviewer_7SFc · 2023-08-12
> >
> > The additional experiments show the effectiveness of the proposed method and the detailed data augmentation description is given. However, I prefer to keep the original decision (borderline accept).

---

### Official Review · Reviewer_VfHM · 2023-07-07

**Soundness:** 2 fair
**Presentation:** 2 fair
**Contribution:** 2 fair
**Rating:** 4
**Confidence:** 4

**Summary:**

This paper proposes a multitask learning framework for video deepfake detection. The idea is to rely on a joint embedding architecture and define a set of coarse-to-fine face forgery detection tasks with corresponding textual descriptions for fake face images (binary level, global-attribute level and local-attribute level). This helps to obtain understandable explanations and hence a more interpretable forensic detector. CLIP is used to implement the joint embedding architecture, while ViT-B/32 is adopted as the visual encoder and GPT-2 as the text encoder. Experiments are carried out on several publicly available datasets and show better performance in terms of generalization compared with SOTA methods.


**Strengths:**

- It is very relevant to design a deepfake detector that is able to generalize to different types of manipulations since often current deepfake methods perform poorly on forgeries not seen during training.
- It is also very valuable to design a detector that can provide explanations about the manipulations.
- The idea to encode the ground-truth labels via language prompts is interesting and not explored yet in the context of deepfakes.


**Weaknesses:**

- The technical description of the method based on multitask learning (Section 3.3) is very generic and not related at all with the problem of deepfakes. In addition, the technical contribution seems to come from already published work: the joint embedding formulation is inspired by minimizing an energy-based model as in [39] and the losses for multitask Learning are inspired by [73].

- The section on Multitask Language Prompts (3.2) is more related to the specific application, but it is not justified why it is important to consider a coarse-to-fine approach and above all it is not clear how the ground-truth labels via language prompts have been generated. It is said 'Face attribute manipulations associated with other textual prompts are already included in FF++.', but this is new to me. FF++ is only labelled using four different manipulations but does not include the global-attribute level and local-attribute level as described in Section 3.2. This is absolutely not clear.

- In this same section there is a reference to a face attribute called 'physical consistency', which is explained in the supplemental material. However, Section 1 of the appendix is very confusing and I was not able to understand it clearly.

- The ablation study is confused. There are several variants that perform well as the proposal in Table 1 and this is puzzling.

- Comparison with SOTA methods should be enlarged including also other methods, such as [66] and [19].

- The experiments that show that the explanations provided by the detector are correct are too limited. They have been shown only on FF++ (Section 4.5 and Appendix). What is more interesting is the ability to generalize to other datasets different from the training one and these are not present in the paper. This is very limiting and does not help to show the relevance of the proposal as stated in the Introduction.

- The paper needs a major re-writing. The presentation is poor and hence not adequate for NeurIPS.

**Questions:**

Please, refer to the weaknesses section.

**Limitations:**

Authors have presented the limitations of their method.

---

> ### Author Rebuttal · Authors · 2023-08-09
>
> **Q1. The technical description of the method based on multitask learning (Section 3.3) is very generic and not related at all with the problem of deepfakes. In addition, the technical contribution seems to come from already published work: the joint embedding formulation is inspired by minimizing an energy-based model as in [39] and the losses for multitask Learning are inspired by [73].**
>
> **A1**: We respectfully disagree with the comment and kindly refer the reviewer to the general response. In short, the most significant contribution is defining a set of coarse-to-fine face forgery detection tasks based on face attributes at different semantic levels. This naturally leads to a multi-task learning setting, which is implemented by a joint embedding approach with several desirable properties regarding semantic encoding, automation, and explainability. The CLIP and the fidelity loss are our instantiations and can be changed to other plausible choices. As pointed out by the reviewer, our approach is generic and can be adapted to other vision problems, which we consider as a huge advantage.
>
> **Q2: The section on Multitask Language Prompts (3.2) is more related to the specific application, but it is not justified why it is important to consider a coarse-to-fine approach and above all it is not clear how the ground-truth labels via language prompts have been generated. It is said 'Face attribute manipulations associated with other textual prompts are already included in FF++.', but this is new to me. FF++ is only labelled using four different manipulations but does not include the global-attribute level and local-attribute level as described in Section 3.2. This is absolutely not clear.**
>
> **A2**:  We have justified experimentally the importance of such a coarse-to-fine approach in Table 3 of the main paper. We have also elaborated on how to encode the ground-truth labels via language prompts through the example in Figure 1 of the main paper. The face image in Figure 1 is a fake face with two attributes altered in different semantic levels, i.e., expression (global) and mouth (local). According to Section 3.2, we have defined nine attributes (including real and fake) for describing the face image in the task of face forgery detection.  If we use “1” to represent the fake label and “0” as the real one, the ground-truth label for the face image in Figure 1 is represented by a one-hot label, i.e., 011000010 (ordered in [real, fake, expression, identity, physical consistency, eye, illumination, mouth, nose]). We further use textual templates to encode the ground-truth label, where each language prompt of nine templates is defined in Section 3.2.  According to the descriptions of each manipulation method in [1, 2, 71, 72] and the notations defined in the FF++ project page, it is straightforward to easily infer the manipulations contained in FF++ from the global-attribute level and local-attribute level.
>
> **Q3. In this same section there is a reference to a face attribute called 'physical consistency', which is explained in the supplemental material. However, Section 1 of the appendix is very confusing and I was not able to understand it clearly.**
>
> **A3**: The concept of physical (in)consistency is not new in the field of photo forensics [R1]. In our case, we create such inconsistency by blending a **real** background with a **fake** foreground (can be the whole face or a face part) generated by a DeepFake algorithm or cropped from another **real** face image of a different ID.  We have explained it in lines 7-16 in Section 1 of the Appendix, with examples of some types of this manipulation in lines 17-20 of Section 1 of the Appendix. To aid understanding, the reviewer thinks of this fake attribute as a form of data augmentation, like the one used in Face X-ray [41] and ICT [19].
>
> [R1] Photo Forensics, Hany Farid, MIT Press.
>
>
> **Q4. The ablation study is confused. There are several variants that perform well as the proposal in Table 1 and this is puzzling.**
>
> **A4**: In the ablation study, we have investigated the impact of different settings of our proposed method and provided the generalization performance on various datasets as evaluation criteria. Although some settings showed reasonable performance, our method with the default setting outperforms them overall. There is only one exception - the use of ViT-B/32 or ViT-L/14 as backbones. We have discussed in the main text why we do not choose them as the default setting in the subsection of the Encoder Architecture of Section 4.4. We kindly request the reviewer to carefully read the explanation in Section 4.4.
>
>
> **Q5. Comparison with SOTA methods should be enlarged including also other methods, such as [66] and [19].**
>
> **A5**: In Table 1 of the main paper, we have already compared our method with many recent state-of-the-art methods, including both [66] and [19], as requested by the reviewer.
> We kindly request the reviewer to carefully read the main text in Section 4.2 and Table 1 of the comparison results with the SOTA.
>
> **Q6. The experiments that show that the explanations provided by the detector are correct are too limited. ... This is very limiting and does not help to show the relevance of the proposal as stated in the Introduction.**
>
> **A6**: The reason for selecting FF++ as the dataset for showcasing examples is that FF++ provides clear descriptions regarding how to use the four included methods to manipulate each image. This provides sufficient grounds for us to relabel the data at both the global-attribute level and local-attribute level. Moreover, using known samples to assess the interpretability of predictions is commonly used in many other literatures [8, 19, 26, 66, 80]. As for the other datasets, which only have global binary annotations, we have comprehensively tested the generalization of our model trained on the FF++ variant in Table 1 of Section 4.2.

---

> > ### Comment · Reviewer_VfHM · 2023-08-16
> >
> > After reading the response from the authors to my comments and to the other reviewers' comments, I increased my score from reject to borderline reject. In fact, authors successfully addressed some of my concerns, however I still believe that the technical contribution is not sufficiently significant for NeurIPS. I also believe that a solution that is proposed for face forgery detection (this is said in the title) should be tailored to this specific task. If it is general, which is considered a plus from the authors, then it should be tested also for other tasks in order to show its relevance also in other applications. Then, note that comparison with [66] and [19] are present in Table 1 but not in Table 2 where robustness is analyzed. Finally, in my opinion the explanations provided by the detector (tested only on FF++) are too limited to prove that the method is working in the correct way. Experiments on other datasets are needed also because the method is trained on this same dataset. Hence under this respect generalization is not verified.

---

### Official Review · Reviewer_jNdm · 2023-07-07

**Soundness:** 2 fair
**Presentation:** 2 fair
**Contribution:** 2 fair
**Rating:** 4
**Confidence:** 3

**Summary:**


The paper appears to be about a method for detecting manipulated facial images, specifically deepfakes. The authors have used a model that employs a joint embedding architecture, using ViT-B/32 as the visual encoder and GPT-2 as the text encoder. The model is trained using AdamW with a decoupled weight decay of 1 × 10−3 and an initial learning rate set to 1 × 10−7, which changes following a cosine annealing schedule. The authors compare their method with several state-of-the-art (SOTA) methods, including Face X-ray, PCL, MADD, LipForensics, RECCE, SBI, ICT, SLADD, and OST. The results show that their proposed method outperforms all the recent SOTA.


**Strengths:**

1.	This paper is well written and easy to follow and will be of interest to researchers from the community of multitask learning in deepfake.

2.	Finding the semantic dependencies among tasks using texture prompts is clear and places the previous work very well in context of this framework.

3.	Finally, the authors provide experimental results to demonstrate the effectiveness of the proposed objective function and algorithms.


**Weaknesses:**

1.	The majority of the contributions in this study are essentially modifications of existing work. Additionally, the significance of the main contribution appears to involve identifying similarities among previous work and proposing a comprehensive generalization that encompasses a significant portion of the existing research. While this contribution may enhance understanding, it seems to be primarily pedagogical in nature rather than being a novel research finding.

2.	The complexity of the proposed method seems high (impractical). How effectively does it handle large datasets? Is it possible to use it in conjunction with sparse variational inference approaches?

3.	It would be great if the authors could extend the proposed algorithm to adapt to other types of loss functions ( from eq.3 – eq.7) such as exp-concave and strongly convex functions.



**Questions:**

1.	How was the drop calculated in Table 2 ?

2.	 Contrastive textual pairing is not defined well?

3.	 How did the minimize energy based model for joint embedding ?

4. What is the performance of the model when dealing with different convex loss functions?

5. How does the energy-based model function within the GPT-2 and CLIP-based embedding framework?

6. What are the challenges and potential solutions when applying your method to face images fully synthesized by GANs or diffusion models?


**Limitations:**

As described in the manuscript, the proposed method may perform unsatisfactorily when encountering fake face images generated by diffusion-model-based methods.

Also, see the weaknesses above.

---

> ### Author Rebuttal · Authors · 2023-08-09
>
> **Q1. The majority of the contributions in this study are essentially modifications of existing work.**
>
> **A1**: We respectfully disagree with the comment and kindly refer the reviewer to the general response. In short, the most significant contribution is defining a set of coarse-to-fine face forgery detection tasks based on face attributes at different semantic levels. This naturally leads to a multi-task learning setting, which is implemented by a joint embedding approach with several desirable properties regarding semantic encoding, automation, and explainability. The CLIP and the fidelity loss are our instantiations and can be changed to other plausible choices.
>
> **Q2. The complexity of the proposed method seems high (impractical).**
>
> **A2:** The reviewer may misunderstand the complexity of our method, which we would like to clarify. The main computational complexity arises from the computation of the image and text embeddings, and the computation complexity of the vision-language correspondence is negligible (i.e., the cosine similarity between image embedding and text embeddings). It is important to note that the text embeddings can be pre-computed only once and used throughout the training and testing procedures. Thus, the main computation complexity comes from extracting the image embedding, which is also required by all competing methods.
>
> **Q3. Regarding the other types of (convex) loss functions.**
>
> **A3:** This paper focuses on the formulation and implementation of multitask learning of face forgery detection at the semantic level, through a novel joint embedding approach. The selection of the best loss function is not our primary focus. Nevertheless, we choose the fidelity loss [R1] over the default cross-entropy loss for several reasons. First, it is capable of obtaining the real minimal loss for each desired probability. Unlike the cross-entropy loss, the fidelity loss has zero loss for each pair, which makes the trained model more accurate. Second, it is bounded between 0 and 1. If the loss has no appropriate upper bound, hard samples continuously placed in the wrong position could lead to excessive loss. In Table 3 of the main paper, we have ablation studies on popular loss functions used in visual tasks that are suitable for our formulation, including the cross-entropy loss, the probabilistic loss, as well as the fidelity loss, and show that the fidelity loss gives the best performance.
>
> [R1] FRank: A ranking method with fidelity loss. In ACM SIGIR, pages 383–390, 2007.
>
> **Q4. How was the drop calculated in Table 2?**
>
> **A4**: The drop is calculated as
>
> $$ \mathrm{Drop}= \frac{\mathrm{mAUC}_{\mathrm{perturb} }-\mathrm{AUC} _{\mathrm{clean} }}{\mathrm{AUC} _{\mathrm{clean} }} \times 100\% $$
>
> where
> $\mathrm{mAUC}_{\mathrm{perturb} }$ is the average of performance on all the perturbations, denoted as Mean AUC in the table, and $\mathrm{AUC} _{\mathrm{clean} }$ is the clean AUC.
>
> **Q5. Contrastive textual pairing is not defined well?**
>
> **A5**: The motivation of the proposed contrastive textual pairing is to encourage the model to learn a more accurate correlation between the visual and textual embeddings via contrastive learning. Specifically, given a fake face with the modified eye(s), the goal is to maximize the similarity between the image and the corresponding textual embedding (i.e., “A photo of a face with the local attribute of {eye} altered” ) and minimize the opposite textual embedding simultaneously. Empirically, we find contrastive textual paring to facilitate model optimization and boost performance, as shown in the following table. As for more design details, we refer the reviewer to Section 3.2 in the main paper.
>
> | Model Variant|CDF|FSh|DF-1.0|DFDC|Mean AUC|
> |-|-|-|-|-|-|
> |Ours (Default)|**89.02**|**98.68**|**93.38**|**82.06**|**90.79**|
> |w/o contrastive textual pairing|87.89|98.34|93.30|81.27|90.20|
>
> **Q6. How did the minimize energy-based model for joint embedding?**
>
> **A6**: The goal of joint embedding is to maximize the vision-language correspondence, while the energy-based model is to minimize the energy of some physical or computational system. In our case, we transfer the vision-language correspondence into the similarity probability, fed into a physically inspired function - the fidelity loss. The fidelity loss for compatible visual and textual embeddings will be low (corresponding to low energy), while incompatible embeddings will lead to a larger loss (corresponding to high energy). Thus, the optimization is consistent with the goal of the energy-based model. As for more design details, we refer the reviewer to lines 162-187 in the main paper.
>
> **Q7. How does the energy-based model function within the GPT-2 and CLIP-based embedding framework?**
>
> **A7**:   In the CLIP model, its text encoder adopts GPT-2 with a base size of 63M-parameter. So, GPT-2 is within the CLIP model. The energy model is built on the similarities (e.g., cosine similarity) between image and textual embeddings. During training, we maximize the similarities between compatible image and textual embeddings and minimize the similarities between incompatible embeddings, which corresponds exactly to energy minimization in machine learning.
>
> **Q8. What are the challenges and potential solutions when applying your method to face images fully synthesized by GANs or diffusion models?**
>
> **A8**: Our current model is mainly trained on the forged faces with blending operations [1, 2, 18, 31, 40, 44, 63, 71, 72], which aims to make the generated face more realistic by alleviating the effect on the authentic region. Thus, it may not perform very well when directly applying it to fully synthesized faces because they do not contain a blending operation. A simple solution is to train our model with the face images fully synthesized by GANs or diffusion models because we do not rely strongly on blending or contrastive features within the fake face image.

---

> > ### Author Response · Authors · 2023-08-18
> > **Follow-up on Review Feedback**
> >
> > We greatly appreciate the time and effort you've invested in reviewing our work and providing constructive feedback. We are following up to check whether our responses have addressed your comments and concerns.
> >
> > Thank you,
> > Authors

---

### Official Review · Reviewer_fx6c · 2023-07-24

**Soundness:** 2 fair
**Presentation:** 3 good
**Contribution:** 2 fair
**Rating:** 5
**Confidence:** 4

**Summary:**

This work proposes an automated multitask learning framework for face forgery detection from a joint embedding perspective. The central idea is to utilize the multi-modality of visual and textural features to enhance blending-based face forgery detection with the global and local semantic face attributes. Experiments demonstrate the effectiveness of this proposed framework.

**Strengths:**

The new paradigm of multitask learning strategy from a joint embedding perspective is introduced into the face forgery detection field. The work trains two encoders to jointly embed visual face images and textual descriptions in the shared feature space. Thus, one can guide the forgery detection that is mainly based on visual content with textural descriptions. This work successfully explored the feasibility of using multi-modality data with a multi-task learning framework. Extensive results on the ablation studies verified the effectiveness of the proposed framework.

**Weaknesses:**

The majority of technical components of this work are borrowed from existing works, e.g., multitask learning, embedding space representation (latent space), textural space and etc. The technical contributions that inspire the following research are quite limited.

**Questions:**

1) The authors suggest using a textual template, where the critical description such as real/fake are instantiated as needed. It is no doubt about the effectiveness of this treatment. However, does it really necessary to handle the textural description using templates? What if directly using the attributes (e.g., real or fake) rather than a textural template (one can see that the differences between the textural descriptions on real images and fake images are only the keyword “real” and “fake”.)
2) For Section 4.3 Robustness Analysis, the considered distortion of perturbations is necessary. However, more distortions such as image/video compression (JPEG or HEVC) are missing. In practical scenarios, the forged images are often communicated via online social networks, which are typically applied compression.
3) The authors stated that the proposed method can only be applied to the scenarios where the forged faces are generated with blending operations. What if the blended image is further processed with some other harmonization techniques (e.g., with illumination correction on the faces or the edges between the beguine and fake regions)
4) The adopted FF++ dataset contains video data for training, validation, and testing. As a well-known fact, one critical information of the faked video is temporal information. However, it seems that the authors neglect the temporal information directly.


**Limitations:**

The authors clearly stated the limitation of this work. The proposed method cannot be applied to totally generated AI-generated images such as GAN-generated or diffusion-based model generated.

---

> ### Author Rebuttal · Authors · 2023-08-09
>
> **Q1. The technical contributions that inspire the following research are quite limited: The majority of technical components of this work are borrowed from existing works, e.g., multitask learning, embedding space representation (latent space), textural space and etc.**
>
> **A1**: We respectfully disagree with the comment and kindly refer the reviewer to the general response. In short, the most significant contribution is defining a set of coarse-to-fine face forgery detection tasks based on face attributes at different semantic levels. This naturally leads to a multi-task learning setting, which is implemented by a joint embedding approach with several desirable properties regarding semantic encoding, automation, and explainability. The CLIP and the fidelity loss are our instantiations and can be changed to other plausible choices.
>
> **Q2. The authors suggest using a textual template, where the critical description such as real/fake are instantiated as needed. It is no doubt about the effectiveness of this treatment.
> However, does it really necessary to handle the textural description using templates? What if directly using the attributes (e.g., real or fake) rather than a textural template
> (one can see that the differences between the textural descriptions on real images and fake images are only the keyword “real” and “fake”.)**
>
> **A2:** As suggested by the reviewer, we conduct additional experiments by 1) simplifying the proposed textual templates to two keywords, real/fake, for textual encoding, 2) representing the two keywords, real/fake, with one-hot labels, followed by MLP encoding, and 3) no text encoder at all (i.e., a ViT-based traditional discriminative architecture that predicts multiple target outputs directly from the input face image). The experimental results in the following table verify the effectiveness of our textual templates.
>
> | Method  | CDF  |  FSh |  DF-1.0 | DFDC  |  Mean AUC |
> | ------------ | ------------ | ------------ | ------------ | ------------ | ------------ |
> | Real/fake textual encoding | 85.91  |  98.45 |  93.34 | 80.44  | 89.54  |
> | One-hot MLP encoding | 84.05  |  98.13 |  92.29 | 79.48  | 88.49  |
> |  No text encoder |  76.25 | 87.37  | 83.24  |  72.89 |  79.94 |
> | Ours  | **89.02**  | **98.68**  |  **93.38**| **82.06** | **90.79**  |
>
>
> **Q3. For Section 4.3 Robustness Analysis, the considered distortion of perturbations is necessary. However, more distortions such as image/video compression (JPEG or HEVC) are missing. In practical scenarios, the forged images are often communicated via online social networks,
> which are typically applied compression.**
>
> **A3:** Thanks for the excellent suggestion. We conduct additional experiments to probe the robustness of the competing detectors to JPEG compression, as shown in the following table. Mean AUC indicates the averaged performance across all perturbations, i.e., Patch-Sub, Noise, Blur, Pixelation, and JPEG Compression. It is clear that the proposed method is capable of maintaining high performance against the distortion of JPEG compression.
>
> | Method  | Clean AUC  |  JPEG Compression | Mean AUC  | Drop |
> | ------------ | ------------ | ------------ | ------------ | ------------ |
> | Face X-ray  |  98.37 |  81.03 |  82.24 | -16.40%  |
> | CNND  | 99.56  |  98.34 | 86.90  | -12.72%  |
> |  LipForensics |  99.90 |  94.64 | 91.30  | -8.61%  |
> | Ours  | 98.49  | 91.91  | 90.08  |  **-8.53%** |
>
> The Drop is calculated as follows,
>
> $$ \mathrm{Drop} = \frac{\mathrm{mAUC}_{\mathrm{perturb} }-\mathrm{AUC} _{\mathrm{clean} }}{\mathrm{AUC} _{\mathrm{clean} }} \times 100\% $$
>
> where
> $\mathrm{mAUC}_{\mathrm{perturb} }$ is the average of performance on all the perturbations, and $\mathrm{AUC} _{\mathrm{clean} }$ is the clean AUC.
>
> **Q4. The authors stated that the proposed method can only be applied to the scenarios where the forged faces are generated with blending operations. What if the blended image is further processed with some other harmonization techniques (e.g., with illumination correction on the faces or the edges between the beguine and fake regions)**
>
> **A4:** The reviewer may misunderstand the blending operation in our face forgery pipeline, which we would like to clarify. After the blending operation, the blended image always undergoes a harmonization process (except for images with inconsistent illumination, for which we do not apply harmonization). Therefore, the proposed model can deal with the scenarios where the blended images are further processed with harmonization techniques. We would like to refer the reviewer to Section 2 and Table 1 in Appendix for more details.
> As pointed out by the reviewer, our model is mainly trained on the forged faces with blending operations, which may not perform well when directly applying it to fully synthesized faces. This problem can be addressed by adding some fully synthesized faces during training.
>
> **Q5. The adopted FF++ dataset contains video data for training, validation, and testing. As a well-known fact, one critical information of the faked video is temporal information. However, it seems that the authors neglect the temporal information directly.**
>
> **A5**: In this paper, we focus on image-based DeepFake detection rather than video-based. Therefore, following many recent methods [11, 19, 66, 81], we do not consider temporal information for a fair comparison.

---

> > ### Comment · Reviewer_fx6c · 2023-08-16
> >
> > Most of my concerns were well addressed, and I would like to upgrade the evaluation to borderline accept.

---

### Official Review · Reviewer_NnkA · 2023-07-26

**Soundness:** 2 fair
**Presentation:** 3 good
**Contribution:** 2 fair
**Rating:** 5
**Confidence:** 5

**Summary:**

This paper introduces a joint embedding approach for multitask learning in face forgery detection. The method defines a set of coarse-to-fine face forgery detection tasks based on face attributes at different semantic levels, and describes the ground truth for each task via a textual template. CLIP is used to implement the joint embedding architecture, and multi-level fidelity losses are used for multitask learning. The proposed method outperforms state-of-the-art detectors in terms of generalization ability.

**Strengths:**

1. This paper proposes a joint-embedding-based multitask learning method for face forgery detection. It could probably be the first work to apply the language prompts on the task of face forgery detection.
2. This paper defines a set of coarse-to-fine face forgery detection tasks based on face attributes at different semantic levels to facilitate the multitask learning.
3. The proposed method achieves better performance than the SOTA schemes in terms of generalization ability.

**Weaknesses:**

1. The authors apply the existing technologies including CLIP and fidelity loss for joint-embedding-based multitask learning. The technical contribution is rather limited.
2. It lacks of explanation of why the authors use CLIP for joint learning. For the same token, it also lacks of analysis regarding the use of fidelity loss for multi-task learning.
3. The works for comparing the robustness do not include the SOTA schemes.



**Questions:**

1. In Table 1, the AUC of OST on DFDC is lower than that is reported in reference [11] (77.73% vs 83.30%), while the AUC of OST on CDF is exactly the same as that is reported in [11]. The authors may want to explain why.
2. The authors may want to explain the purpose of reporting the performance of w/o Aug  in Table 2 and 3.
3. In section 1, lines 71-72, “textural templates” should be “textual templates”.


**Limitations:**

Please refer to weaknesses.

---

> ### Author Rebuttal · Authors · 2023-08-09
>
> **Q1. Regarding the limited technical contribution. The authors apply the existing technologies, including CLIP and fidelity loss for joint-embedding-based multitask learning.**
>
> **A1**: Please refer to the general response for technical contributions. In short, the most significant contribution is defining a set of coarse-to-fine face forgery detection tasks based on face attributes at different semantic levels. This naturally leads to a multi-task learning setting, which is implemented by a joint embedding approach with several desirable properties regarding semantic encoding, automation, and explainability. The CLIP and the fidelity loss are our instantiations and can be changed to other plausible choices.
>
>
> **Q2. It lacks of explanation of why the authors use CLIP for joint learning. For the same token, it also lacks of analysis regarding the use of fidelity loss for multi-task learning.**
>
> **A2**: As CLIP is a simple yet prevalent vision-language model, we use CLIP to compute text/image embeddings. A significant advantage of encoding ground-truth labels via textual prompts is that it gives us a great opportunity to leverage the semantic dependencies among tasks in the representation space. It is also possible to explore embeddings from other vision-language models like UniCL, LiT, GroupViT, HiCLIP, etc.
> For the fidelity loss, it has the following advantages over the cross-entropy loss. 1) It is capable of obtaining the real minimal loss for each desired probability. Unlike the cross-entropy loss, the fidelity loss has zero loss for each pair, which makes the trained model more accurate. 2) It is bounded between 0 and 1. If the loss has no appropriate upper bound, hard samples continuously placed in the wrong position could lead to excessive loss. This can bias the model and degrade its performance. We also experimentally demonstrate the superiority in our ablation studies.
>
> **Q3. The works for comparing the robustness do not include the SOTA schemes.**
>
> **A3**: The primary goal of this paper is to improve model generalizability rather than robustness. Thus, our experimental setups are mainly designed for a fair comparison and testing of model generalizability. In the robustness testing, we include several SOTA schemes of Face X-ray, CNND, and Lip-forensics because the former two also use data augmentation during training, and the latter relies on high-level semantic features with intrinsic robustness to low-level manipulations. As suggested by the reviewer, we will incorporate other SOTA schemes for the robustness comparison.
>
> **Q4. In Table 1, the AUC of OST on DFDC is lower than that is reported in reference [11] (77.73% vs 83.30%), while the AUC of OST on CDF is exactly the same as that is reported in [11]. The authors may want to explain why.**
>
> **A4**: There are two tables in OST; one (Table 1 in OST) is for generalizability comparison, and the other (Table 2 in OST) is for comparison with models based on meta-learning.
> The AUC of OST on DFDC in this paper is the average result (the calculation is in line with that in the OST paper) of what is reported in Table 1 of the OST paper, while the AUC of OST on CDF is directly copied from the result in Table 2 of OST
> because there are no other results reported in the original OST paper. As suggested by the reviewer, we report the AUC of OST on DFDC based on Table 2 in the OST paper, as follows. From the table, the proposed method still outperforms OST in terms of five face forgery datasets by a clear margin.
>
> | Method  | FF++  |  CDF | FSh  |  DF-1.0 | DFDC  | Mean AUC  |
> | ------------ | ------------ | ------------ | ------------ | ------------ | ------------ | ------------ |
> | OST  | 98.20  |  74.80 |  -- |  93.08 | **83.30**  | 87.34/83.73  |
> |  Ours | **98.49** | **89.02**  | 98.68  |  **93.38** |  82.06 |  **92.33**/**90.79** |
>
> **Q5. The authors may want to explain the purpose of reporting the performance of w/o Aug in Table 2 and 3**.
>
> **A5:** Existing face forgery detection methods tend to use very different data augmentation strategies for performance boosting. We report the performance of the proposed model w/o data augmentation in Tables 2 and 3 with the goal of singling out the core contribution of our approach: multitask learning of face forgery detection via joint embedding.
>
> **Q6. A typo: In section 1, lines 71-72, “textural templates” should be “textual templates”.**
>
> **A6**: Thanks for pointing out this typo; we will revise it and proofread the whole manuscript.

---

> > ### Comment · Reviewer_NnkA · 2023-08-17
> >
> > I maintain my initial rating after reading the response and other reviewers' comments.

---

### Author Rebuttal · Authors · 2023-08-09

### **A general response regarding the contributions of our work**

We thank all reviewers for the detailed and constructive comments. We are glad to find that most reviewers generally acknowledge the following contributions of our work.

This paper explores multitask learning of face forgery detection from a joint embedding perspective, aiming to improve generalizability and explainability.

As highlighted by ***Reviewer VfHM***, it is valuable to design a detector that can provide explanations about the manipulations;

As highlighted by ***Reviewer NnkA*** and ***fx6c***, this paper is a pioneering effort in employing language prompts or multimodal data to address the challenge of face forgery detection, which shed light on multimodal approaches for face forgery detection;

As highlighted by ***Reviewer jNdm***, this paper is easy to follow and will be of interest to researchers from the community of multitask learning in DeepFake.

We would like to emphasize that our approach is not a simple combination of existing techniques but with solid motivations and justifications.

$\underline{\text{First}}$, unlike most existing methods, we prefer to tackle face forgery detection at the semantic level rather than at the signal level. To achieve this, we have defined a set of coarse-to-fine face forgery detection tasks based on face attributes at different semantic levels. This naturally leads to a multitask learning formulation of face forgery detection.

$\underline{\text{Second}}$, the prevailing multitask learning paradigm for face forgery detection takes a discriminative approach, i.e., predicting multiple target outputs (one for each task) directly from the input face image. Such a paradigm suffers from two main drawbacks.
**1)** It overlooks semantic relationships across tasks, which weakens knowledge transfer. For example, irrelevant information (e.g., every detail of the face image in face reconstruction [8]) may be transferred across tasks.
**2)** It requires extensive human expertise to determine task-agnostic/task-specific model parameters and the weights of different task losses as two forms of hyperparameters.

As a significant departure, we propose to formulate multitask learning using a novel joint embedding paradigm. This paradigm is capable of directly transferring the recent advances in multimodal learning (in particular, text + image), which 1) supports encoding the semantic closeness between tasks in the latent feature space, 2) enables automated multitask learning in terms of allocating model capacity (i.e., specifying task-agnostic and task-specific model parameters) and 3) provides textual explanations. All these have not been accomplished by previous methods. In addition, this paradigm takes initial steps and sheds light on face forgery detection using multimodal information.

---

### Decision · Program_Chairs · 2023-09-21

**Decision:**

Reject

**Comment:**

The paper introduces a multitask learning framework for detecting face forgery, which leverages both visual and textual data by utilizing CLIP. The framework defines a hierarchy of face forgery detection tasks, facilitating automated multi-task learning and interpretability. It demonstrates the effectiveness on deepfake datasets and provides explanatory sentences for authenticity verification, enhancing its value in deepfake detection.

However, it's worth noting that this paper has received mixed ratings from reviewers. A major concern among reviewers is the perceived limited technical contribution. Some reviewers were initially skeptical, and thanks to further clarification in the rebuttal. One significant point raised by Reviewer VfHM, about tailoring the solution more specifically to face forgery detection, should be carefully considered. The authors' response to this concern does not seem to have convinced Reviewer VfHM and the AC. The AC emphasizes the importance of addressing this concern, as it suggests that further work is needed to optimize the proposed solution for face forgery detection. The performance of the proposed approach, as demonstrated in Table 1, falls short of the SOTA, particularly on the CDF and DF-1.0 datasets. The use of Mean AUC is also questioned since it should ideally demonstrate consistent improvement across all datasets to showcase its generalization and robustness in diverse face forgery detection scenarios. Additional reviewers have also raised questions about the robustness of the approach, a crucial aspect in face forgery detection. The AC underscores the need for further improvements, such as leveraging a pretrained CLIP model on face-related datasets to enhance the discrimination of facial attributes. Additionally, investigating the sensitivity of the template and template size through ablation studies is suggested.

In conclusion, the AC strongly encourages the authors to undertake additional work to enhance the quality of the paper and calls for a new round of reviewing. At present, the paper is not recommended for acceptance or conditional acceptance.